# Range of motion and between-measurement variation of spinal kinematics in sound horses at trot on the straight line and on the lunge

A. M. Hardeman[1,2,3]*, A. Byström[3], L. Roepstorff[3], J. H. Swagemakers[1], P. R. van Weeren[2], F. M. Serra Bragança[2]

**1** Tierklinik Luesche GmbH, Luesche, Germany, **2** Department of Clinical Sciences, Faculty of Veterinary Medicine, Utrecht University, Utrecht, The Netherlands, **3** Department of Anatomy, Physiology and Biochemistry, Swedish University of Agricultural Sciences, Uppsala, Sweden

* aagje.hardeman@gmail.com

**Data Availability Statement:** All relevant data are within the manuscript and its Supporting Information files.

## Abstract

Clinical assessment of spinal motion in horses is part of many routine clinical exams but remains highly subjective. A prerequisite for the quantification of spinal motion is the assessment of the expected normal range of motion and variability of back kinematics. The aim of this study was to objectively quantify spinal kinematics and between -measurement, -surface and -day variation in owner-sound horses. In an observational study, twelve owner-sound horses were trotted 12 times on four different paths (hard/soft straight line, soft lunge left and right). Measurements were divided over three days, with five repetitions on day one and two, and two repetitions on day three (recheck) which occurred 28–55 days later. Optical motion capture was used to collect kinematic data. Elements of the outcome were: 1) Ranges of Motion (ROM) with confidence intervals per path and surface, 2) a variability model to calculate between-measurement variation and test the effect of time, surface and path, 3) intraclass correlation coefficients (ICC) to determine repeatability. ROM was lowest on the hard straight line. Cervical lateral bending was doubled on the left compared to the right lunge. Mean variation for the flexion-extension and lateral bending of the whole back were 0.8 and 1 degrees. Pelvic motion showed a variation of 1.0 (pitch), 0.7 (yaw) and 1.3 (roll) degrees. For these five parameters, a tendency for more variation on the hard surface and reduced variation with increased repetitions was observed. More variation was seen on the recheck (p<0.001). ICC values for pelvic rotations were between 0.76 and 0.93, for the whole back flexion-extension and lateral bending between 0.51 and 0.91. Between-horse variation was substantially higher than within-horse variation. In conclusion, ROM and variation in spinal biomechanics are horse-specific and small, necessitating individual analysis and making subjective and objective clinical assessment of spinal kinematics challenging.

## Introduction

Back pain/dysfunction is a common cause of poor performance in horses [1,2] which can cause alterations in spinal kinematics [3,4]. However, apart from a primary back problem,

**Funding:** Tierklinik Lüsche GmbH' provided support in the form of salaries for authors [JH (fourth author), AH (first author)], but did not have any additional role in the study design, data collection and analysis, decision to publish, or preparation of the manuscript that could lead to conflicting situations. The specific roles of these authors are articulated in the 'author contributions' section.

**Competing interests:** The involvement of 'Tierklinik Lüsche GmbH' does not alter our adherence to PLOS ONE policies on sharing data and materials.

lameness may also affect spinal biomechanics, as was shown in studies on the effects of induced lameness[5,6]. The rider may experience consequences of back dysfunction of the horse, either by the reluctance of the horse to bend, sidedness or abnormal saddle movement. These associations are complex [7–9].

At trot, the locomotion pattern can be described as a two-beat, symmetric, diagonal gait with a suspension phase. This creates a bouncing movement, resulting in a sinusoidal pattern for head, withers and tuber sacrale[10]. A similar sinusoidal pattern is observed for the flexion-extension of the back, with one cycle per diagonal. At the trot, movements of the back are mainly a result of forces applied to the spine by the fore- and hindlimbs and the weight of the abdominal viscera. Both the epaxial and hypaxial muscles play an important role in controlling the flexion-extension and thereby stabilizing the spine. The abdominal muscles, which act to flex the back, are mainly active during the impact phase of the stride, when the back is going to extension. Correspondingly, the back extensors, the epaxial muscles, are active during the push off half of the stride when the back is going to flexion[11]. Whereas the spine undergoes two cycles of flexion-extension per stride, there is only a single cycle of lateral bending and axial rotation during one stride at the trot. Lateral bending, in the horizontal plane, is seen ones to the left and ones to the right side during one stride-cycle. The same is true for axial rotation (around the longitudinal axis). Not much is known until now about the changes in back motion on the lunge, other than a higher ROM compared to the straight line[12]. The clinical diagnosis of back pain/dysfunction in horses is quite challenging. Additional diagnostic tools, besides a proper anamnesis and a complete clinical examination, such as scintigraphy, radiology and ultrasonography are therefore frequently employed to maximize evidence, but oftentimes the outcome is still far from conclusive and false positive results are common [11,13]. For this reason, an objective tool to evaluate back motion would be a useful asset in the clinical situation. First off, because changes in spinal kinematics are subtle and hence difficult to visually assess [14,15]. Secondly, it is well-known that subjective assessment of equine lameness is characterized by high inter-observer variability and strongly susceptible to bias [16,17]. The unreliability of subjective evaluation of spinal kinematics is likely to be only greater compared to lameness assessment, given the generally much subtle changes in ROM (before versus after intervention) than in cases of lameness. For the correct clinical interpretation of objective and quantitative data on equine spinal kinematics it is paramount to first quantify normal ranges of motion (ROM) and to evaluate the expected normal amount of biological variation. For frequently used lameness parameters, normal variation has already been addressed [18–20]. Previous work on the normal variation in back kinematics achieved a high repeatability through standardization of the protocol and the use of treadmill locomotion. More variation was found between versus within horses [14]. Back kinematics captured on a treadmill in horses with back dysfunction [3] have been compared to kinematics of a group of asymptomatic horses [21]. There were some significant, but rather small differences in back ROM between the groups. Variation in spinal kinematics in the over-ground situation and on different paths and surfaces, as encountered in the clinical situation, has not been investigated yet.

The aim of the study was to establish normal ROMs in spinal kinematics in clinically sound horses trotting over-ground, including the quantification of the variation between horses and within horses over time. These data may serve as guidelines when interpreting biomechanical changes after an intervention, such as manipulation, medication, training or shoeing.

We hypothesized that between-horse variation would be larger compared to within-horse variation and that between-day variation would be larger than within-day variation.

## Material and methods

Data collection took place in Germany. According to German law and regulations, ethical approval is not required for non-invasive experiments where animals are not subjected to any additional risks related to the study, outside normal handling. Thus, no ethical permission was required for this study. Informed consent for data collection was obtained from the horse owners prior to the study.

### Horses

A detailed description of the study population has been published previously [18]. In brief, 12 sports horses in regular work (three geldings and nine mares) with a body mass range of 450–652 kg (mean 551 kg) and an age range of 5–15 years (mean 8.3 years) were used. Eleven horses were European warmbloods and one was a Friesian. Their competition level varied from not competing until intermediate level in either jumping or dressage. The horses were in regular use, deemed sound by their owner or rider and did not have any history of back or neck problems. An experienced equine veterinarian examined the horses on the day before the first measurement and graded them as sound or close to sound ('fit to compete') defined as less than 1 on the 0 to 5 AAEP lameness scale [22]. This judgment was based on a subjective assessment of a straight-line trot up on a soft surface (hard surface was not available at that timepoint).

### Marker placement

Each horse was equipped with spherical reflective markers (soft spherical marker, 25 mm diameter [a]), attached to the skin with double-sided adhesive tape. The location of each marker was identified by clipping a small proportion of hair to ensure exact replacement of markers on the following days.

Three markers were placed in the frontal plane of the head (the lowest marker was used as the reference marker) and three markers on the withers (one on the highest point, two markers 20 cm lateral to the central one. A T-shaped strip with one marker at each end, was placed so that the three markers were located at the tuber sacrale and the craniodorsal aspect of both tubera coxae. Additional markers were attached to the skin above the dorsal spinous processes of T12, T15, T18, L3, L5 and the sacrum (S5). Marker placement is illustrated in Fig 1. Position was defined by palpation by the same researcher (AH) for all horses.

### Data collection

Optical motion capture data were recorded using Qualisys Motion Capture software (QTM[a] version: 2.14, build: 3180), connected to 28 high-speed infrared cameras (Oqus 700+[a]) set to a sampling frequency of 100Hz. The total covered area in this set-up was approximately 250 m$^2$, height covered was at least 5 m. Calibration was done daily before the start of the measurements, according to the manufacturer's instructions. The average calibration residual was 3.2 mm. Video recordings, synchronized with the motion capture system, were obtained for each measurement (Sony HDR-CX330).

### Measuring protocol

The horses were divided into two groups for logistical reasons but subjected to identical measuring protocols. Measurements were repeated on 12 occasions over a period of up to 55 days. For each horse, measurements were grouped as five replicates on the first and five replicates on the second measurement day, followed by two replicates on the third measurement day

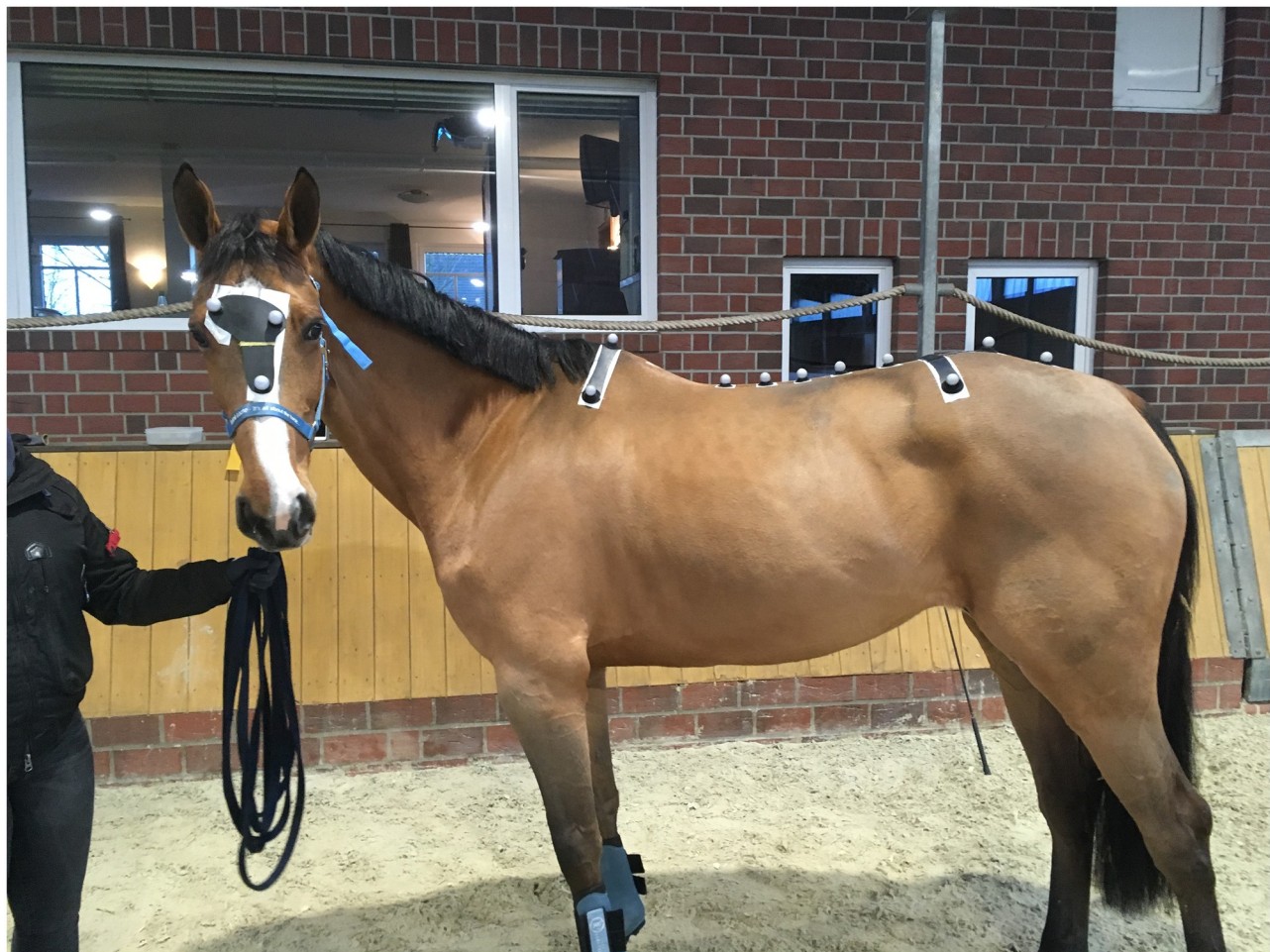

**Fig 1. Marker placement in one of the study subjects.**

(recheck). Between measurement days two and three, there was a period without measurements of at least 28 days. The time schedule of the data collection for each horse can be found in S1 Table.

Each measurement day started with a warm-up period of five minutes hand walking and ten minutes lunging. After the warm-up up period, markers were placed. Measurements were then performed with a five-minute interval between the first two measurements of each day (M1-M2, M6-M7, M11-M12) and with ten minutes in between the remaining measurements of that day (M2-M3-M4-M5, M7-M8-M9-M10). The sequence of registrations was hard (tarmac) straight line (2x20 m), soft straight line (2x30 m), and left and right lunge on soft surface (diameter approximately 10 m, length of lunge-line standardized by a knot), for all measurements (M1-M12). On the lunge, horses were measured for 25 s in each direction. The soft surface consisted of a combination of sand and synthetic fiber, which was harrowed daily before the first measuring session. Horses were trotted at their own preferred speed. Care was taken to minimize changes in speed, ensuring a steady-state movement during the whole measurement. The same handler always handled all horses in a group.

After each measurement, the 3D tracked data were visually inspected ensuring that all markers had been tracked adequately and data were suitable for analysis. Measurements with

poor marker tracking or insufficient number of collected strides (five or less complete strides) were discarded.

## Kinematic data analysis

Table 1 gives an overview of all analysed variables. Kinematic data were analysed using custom-made Matlab scripts [c]. Filtering of the data was performed as previously described [23], and further for all the back segments data, a 4th order low pass Butterworth filter with a cut-off frequency set to 30Hz was used to remove the high frequency noise present in the data.

Stride segmentation was done as earlier described [18]. Speed was calculated by smoothed differentiation of the horizontal coordinates (x, y) of the marker on the tuber sacrale.

'Whole back flexion-extension' and 'Whole back lateral bending' were calculated as the angle between the two segments 'withers—T15' and 'T15—tuber sacrale', in the sagittal plane for flexion-extension and in the dorsal (horizontal) plane for lateral bending. Segment angles (T12, T15, T18, L3, L5 and tuber sacrale) were calculated in the same way as the flexion-extension and lateral bending of the whole back, using the markers cranial and caudal to the vertebra/marker in question. For example, the flexion-extension of T15 was calculated as the angle

**Table 1. Kinematic parameters; 5% and 95% percentiles and median value, per parameter.**

| Variable | Units | Hard straight | | | Soft straight | | | Soft left | | | Soft right | | |
|---|---|---|---|---|---|---|---|---|---|---|---|---|---|
| | | 5% | median | 95% | 5% | median | 95% | 5% | median | 95% | 5% | median | 95% |
| **Stride duration** | sec | 0.71 | 0.73 | 0.78 | 0.70 | 0.75 | 0.79 | 0.72 | 0.79 | 0.82 | 0.73 | 0.78 | 0.83 |
| **Stride frequency** | Hz | 1.26 | 1.41 | 1.50 | 1.25 | 1.34 | 1.53 | 1.17 | 1.25 | 1.44 | 1.21 | 1.30 | 1.51 |
| **Speed** | m/s | 3.13 | 3.35 | 3.78 | 3.26 | 3.79 | 4.39 | 2.97 | 3.30 | 3.74 | 3.04 | 3.35 | 3.63 |
| **Head ROM** | mm | 51.99 | 67.21 | 87.75 | 52.56 | 70.89 | 100.09 | 58.05 | 87.10 | 125.38 | 63.12 | 94.95 | 144.50 |
| **Withers ROM** | mm | 66.02 | 86.19 | 99.03 | 72.30 | 93.56 | 106.13 | 86.43 | 109.58 | 131.62 | 83.35 | 106.33 | 127.91 |
| **Sacrum ROM** | mm | 76.74 | 90.01 | 95.42 | 80.54 | 96.34 | 106.14 | 84.15 | 106.79 | 129.05 | 88.33 | 110.21 | 125.12 |
| **Pelvis roll (AR)** | deg | 5.84 | 8.51 | 9.08 | 6.37 | 9.65 | 12.94 | 5.90 | 8.77 | 14.09 | 6.09 | 9.35 | 12.87 |
| **Pelvis pitch (flex-ext)** | deg | 4.92 | 6.98 | 8.16 | 5.01 | 7.62 | 10.50 | 6.06 | 8.08 | 11.09 | 5.97 | 8.27 | 11.34 |
| **Pelvis yaw (lat bend)** | deg | 3.12 | 3.95 | 4.94 | 3.07 | 4.29 | 6.58 | 3.89 | 5.23 | 6.53 | 3.87 | 5.26 | 7.11 |
| **Body tracking** | deg | -1.60 | 0.61 | 1.78 | -2.17 | 0.27 | 2.59 | -1.01 | 2.31 | 6.72 | -4.82 | -2.67 | -0.51 |
| **Head swivel** | deg | -8.89 | -0.40 | 2.65 | -6.52 | -1.28 | 4.00 | -17.09 | -7.93 | 4.27 | -5.70 | 4.48 | 13.90 |
| **Whole back flex-ext** | deg | 3.98 | 4.91 | 5.61 | 4.00 | 4.97 | 6.27 | 4.38 | 5.71 | 7.12 | 4.19 | 5.55 | 6.66 |
| **Whole back lat bend** | deg | 4.96 | 6.46 | 8.08 | 5.10 | 7.45 | 11.22 | 6.47 | 7.80 | 10.86 | 5.69 | 7.77 | 10.89 |
| **Flexion-extension T12** | deg | 2.45 | 3.42 | 5.83 | 2.68 | 3.77 | 6.53 | 2.63 | 3.94 | 7.20 | 2.62 | 3.88 | 8.55 |
| **Lateral bending T12** | deg | 5.52 | 7.24 | 14.88 | 4.73 | 7.56 | 19.26 | 5.18 | 7.42 | 16.67 | 4.49 | 7.35 | 15.13 |
| **Flexion-extension T15** | deg | 1.79 | 2.08 | 3.87 | 1.64 | 2.10 | 3.45 | 1.47 | 2.03 | 3.88 | 1.49 | 1.98 | 4.30 |
| **Lateral bending T15** | deg | 3.47 | 4.89 | 10.00 | 2.96 | 4.62 | 8.01 | 3.25 | 4.14 | 9.00 | 2.99 | 4.33 | 8.44 |
| **Flexion-extension T18** | deg | 1.73 | 1.92 | 3.07 | 1.40 | 2.29 | 2.99 | 1.38 | 2.31 | 2.91 | 1.86 | 2.55 | 3.29 |
| **Lateral bending T18** | deg | 1.88 | 3.20 | 8.07 | 1.87 | 3.10 | 7.21 | 2.26 | 3.26 | 8.46 | 2.11 | 3.38 | 8.26 |
| **Flexion-extension L3** | deg | 1.83 | 2.38 | 3.75 | 1.72 | 2.33 | 3.60 | 1.84 | 2.35 | 3.62 | 1.61 | 2.50 | 3.59 |
| **Lateral bending L3** | deg | 2.99 | 3.35 | 5.31 | 2.83 | 3.79 | 5.54 | 2.99 | 4.13 | 5.61 | 2.82 | 4.03 | 6.36 |
| **Flexion-extension L5** | deg | 1.30 | 2.91 | 4.41 | 1.57 | 2.30 | 4.65 | 1.81 | 2.56 | 4.86 | 1.81 | 2.60 | 5.08 |
| **Lateral bending L5** | deg | 2.36 | 3.07 | 5.07 | 2.76 | 4.22 | 8.11 | 2.69 | 4.20 | 6.78 | 2.96 | 4.43 | 7.15 |
| **Flex-ext tuber sacrale** | deg | 2.61 | 3.46 | 3.83 | 2.67 | 3.62 | 4.86 | 2.82 | 3.86 | 5.05 | 2.91 | 3.63 | 4.83 |
| **Lat bend tuber sacrale** | deg | 2.71 | 3.75 | 5.14 | 3.32 | 4.63 | 6.79 | 3.32 | 4.80 | 6.30 | 3.53 | 4.64 | 6.52 |

Values are calculated over all 12 horses and all available repetitions per horse for each path and surface combination, using measurement mean values. 'ROM' = Range of Motion, 'flex-ext' = Flexion-extension, 'lat bend' = Lateral bending, 'AR' = Axial rotation, 'T' = thoracic and 'L' = lumbar.

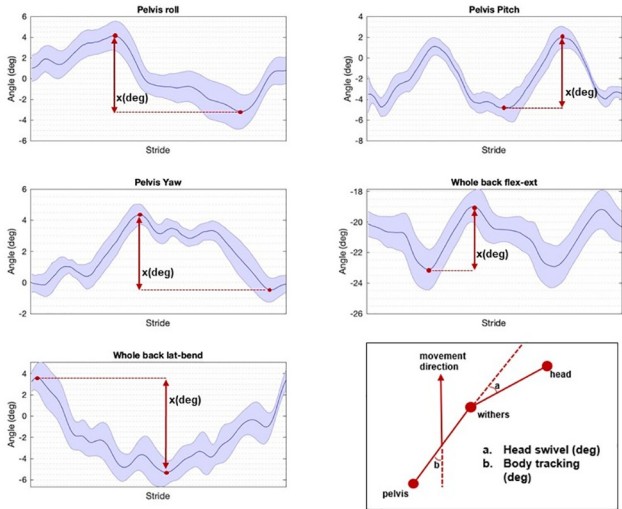

**Fig 2. Mean stride for the 5 main parameters of one horse, on the soft straight line: 'Pelvis roll', 'Pelvis pitch', 'Pelvis yaw', 'Whole Back Flexion-extension' and 'Whole Back Lateral bending'.** On the right bottom, (a) 'Head swivel' and (b) 'Body tracking' are illustrated (degrees). Blue line indicates the mean, shaded area the standard deviation. 'X' illustrates how the values for the different parameters were calculated (degrees). 'AR' = Axial rotation, 'flex-ext' = Flexion-extension, 'lat bend' = Lateral bending.

between T12-T15 and T15-T18, in the sagittal plane. To avoid projection errors, planes were corrected for the horse body lean angle, determined as stride mean pelvic roll during one complete stride [24]. Pelvic roll (axial rotation), pitch (flexion-extension) and yaw (lateral bending), all illustrated in [25], were calculated as projection angles in the frontal, sagittal and dorsal planes, respectively, using data from markers at the tuber sacrale and both tubera coxae. Calculations are illustrated in Fig 2.

The straightness of the body relative to the direction of motion (body tracking) was calculated as the angle in the horizontal plane between the direction of the body (withers to pelvis) and the body velocity vector (direction of movement). Similarly, the head swivel estimates the amount of cervical lateral bending and was calculated as the angle between the cervical spine (head to withers) and the body (withers to pelvis) (Fig 2). For body tracking, a positive value indicates tracking of the forehand to the right and the hind quarters to the left. For head swivel, a positive value indicates cervical bending to the right.

## Statistical analysis

Open software R (3.3.1) [b] was used for statistical analysis. Three different statistical analyses were performed:

1. 5%, 50% (median) and 95% percentiles were determined for each of the different path and surface combinations for all parameters, i.e. back angles, pelvic rotations, body tracking, head swivel and speed. This was done over all 12 horses and all available repetitions, using measurement mean values.

2. Mixed models ('Variability Model') were used to address between-measurement variation. This was done by creating an 'offset adjusted' dataset. First, measurement means were calculated over all available strides. Then the mean of all measurements (M1-M12) for each horse, path and surface combination was subtracted, thus data for each horse were centered around zero per path-surface combination. Absolute values of the 'offset adjusted' dataset

were used as outcome (dependent) data, and were square root transformed due to skewness of the model residuals. Fixed effects were day (day one, day two and recheck), measurement number (day one (1–5), day two (6–10) and 11–12 at recheck), path and surface. Horse ID was used as a random effect. Significance was set at $p < 0.05$. Speed was added to each model as a linear effect. If significant, model estimates from the models with and without speed were compared, to evaluate the influence of speed in the outcome variables. Interactions between fixed effects could not be evaluated (because of no measurements of circles on hard surface) and models were not reduced. Prediction intervals (95%) were calculated for each path and surface combination. As data were offset-adjusted (zero-centered) and prediction intervals thus symmetric around zero, only the upper limits have been tabulated.

The R packages dplyr, lme4, lmerTest, lsmeans, psychometric and ggplot2 were used. Normality of the model residuals was checked using q-q plots and box-plots and homoscedasticity was checked by plotting the fitted values versus the residuals.

1. To address the repeatability of the different parameters, the intra-class correlation coefficient (ICC) was calculated for each path-surface combination with the R function ICC.lme (version v 2.2) using the horse, surface and path (straight line or circle) as grouping variables, using measurement mean values (non-offset adjusted).

## Results

Three horses (horses 3, 8, 10) were not available for the last measuring session (M11-12). One measurement was lost due to technical issues (horse two, M2, soft left circle). Due to marker misplacement (T12 and T15) of horse two, the recheck measurements (M11 and M12) were discarded. A total of 482 measurements were used, 61 were discarded because of less than five strides (hard straight line). All data used in the analysis and for the graphics is available in S4 Table.

For the straight-line measurements, the mean (s.d.) number of included strides per measurement was 14 (3.8). For the lunge, the number of strides per measurement was 36.8 (5.6) and mean circle diameter was 9.7 (0.6) m (based on the trajectory of the tuber sacrale marker). The baseline values for the typical lameness parameters of each horse can be found elsewhere [18]; none of the horses had a lameness score higher than the chosen threshold of 1 out of 5 on the AAEP scale at any of the study days. Therefore, none of them was excluded from the study.

### Quantification of range of motion

Table 1 presents ROM of all parameters, 5% and 95% percentiles and median values, calculated over all 12 horses and all 12repetitions, except for the excluded data as mentioned above. For all back and pelvic parameters, the lowest median values were obtained on the hard straight line. Higher ROMs in four of the five main parameters (Figs 3–7) were seen on the lunge compared to the straight line. On the left lunge, the head swivel angle was twice as large as on the right lunge.

### Variation between and within horses

Variation between and within horses, and between and within days (absolute difference from the mean of all repetitions) of the five main parameters is visualised in Figs 3–7, and of the back segments in S2 Table. Variation between versus within horses is further visualised as box-plots in Fig 8 and S1 Fig, with fairly small individual boxes compared to more considerable differences between the different horses.

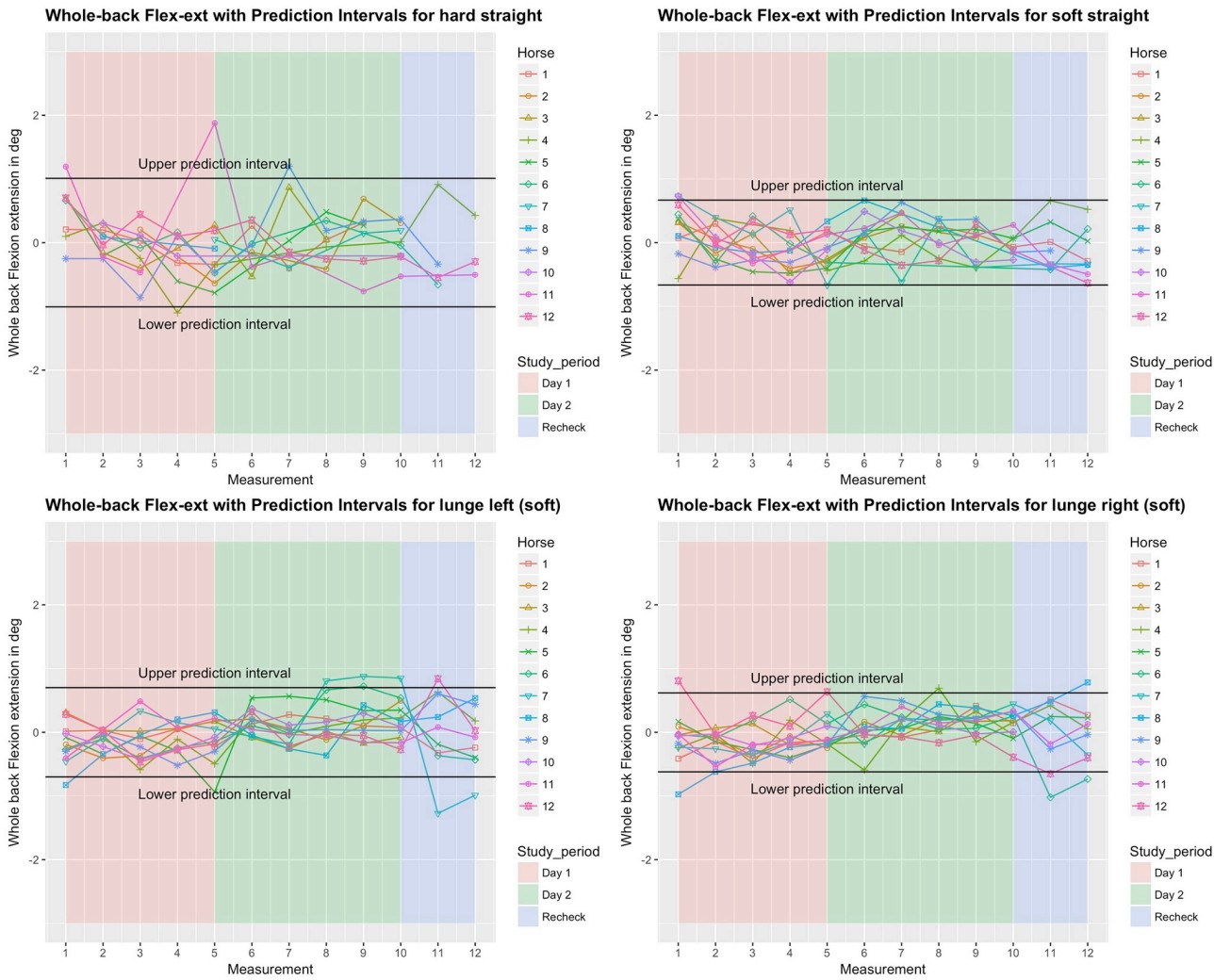

**Fig 3. Between-measurement variation (offset adjusted data) for 'Whole Back Flexion-extension' (calculated as the angle between the two segments 'withers—T15' and 'T15—tuber sacrale'), per measurement, per day and per horse (measurement-mean data).** Black lines indicate 95% prediction intervals.

Prediction intervals for the between-measurement variation of all parameters can be found in Table 2, and for the back segments in S2 Table. Mean prediction intervals (average over the four path-surface combinations) for flexion-extension and lateral bending of the whole back were (±) 0.8 and 1.0 degree, respectively and for pelvic pitch, yaw and roll 1.0, 0.7 and 1.3 degrees, respectively. The mean prediction interval for speed was 0.4 m/s, with a maximum of 0.6 m/s on the hard straight line. Mean prediction intervals for the back segments varied between 0.6 and 1.2 degrees (S2 Table).

### Effect of time, surface and path on the variation

In the variability model, for all five main parameters (Figs 3–7), between-measurement variation (absolute difference from the mean of all repetitions) tended to reduce over repetitions (S3 Table). There was generally more variation on the hard straight line. Significantly more variation was observed at the recheck (p<0.001). Significantly more variation at the recheck was also observed for the variable speed (p = 0.04), but it did not have a significant effect on all

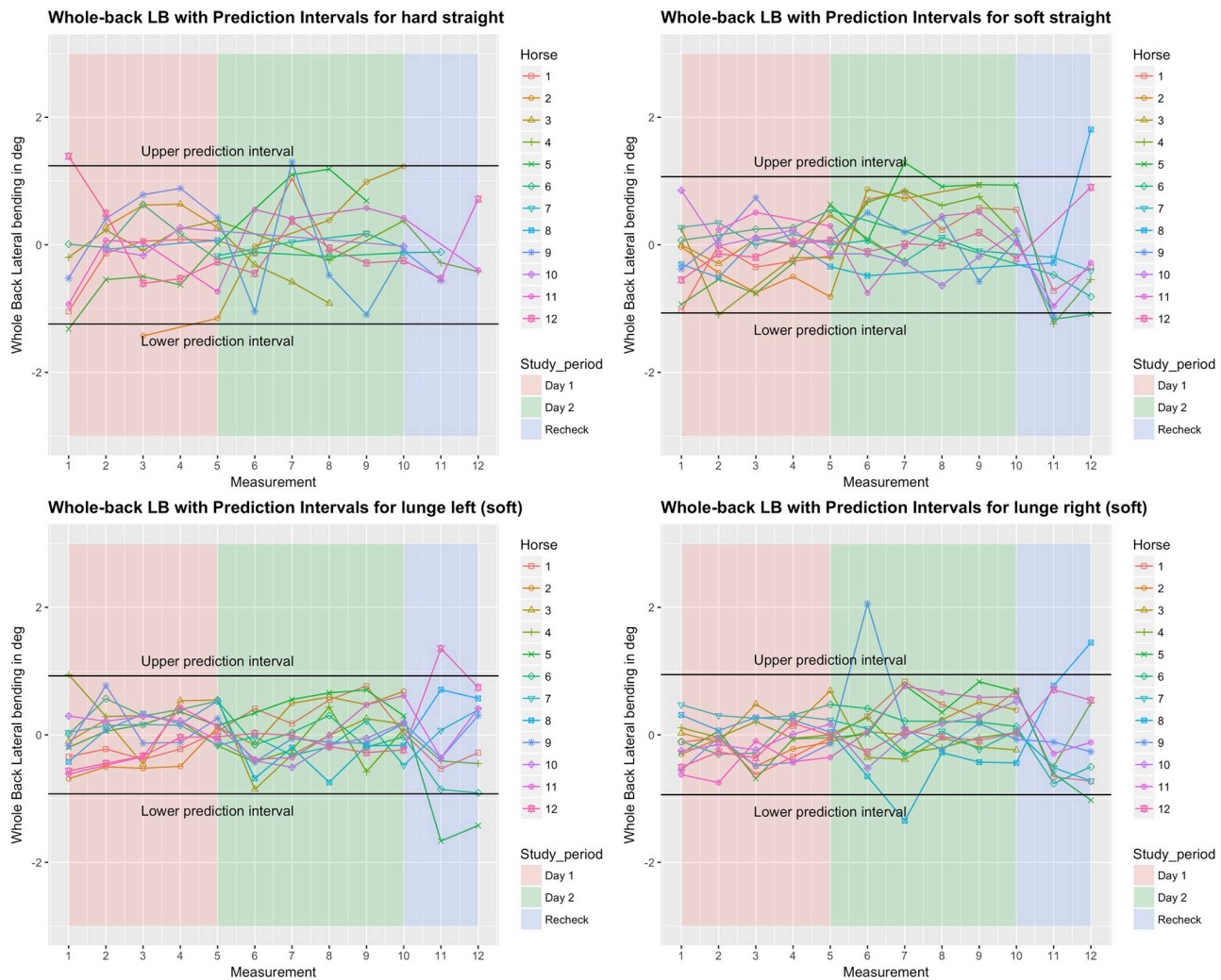

**Fig 4. Between-measurement variation (offset adjusted data) for 'Whole Back Lateral bending' (calculated as the angle between the two segments 'withers—T15' and 'T15—tuber sacrale'), per measurement, per day and per horse (measurement-mean data).** Black lines indicate 95% prediction intervals.

five main parameters when speed was added to the model. Only for pelvic yaw, there was a tendency of speed to have an effect on the model outcomes (p = 0.08) with a positive estimate, but adding speed had only marginal influence on the other estimates.

Head swivel angle showed the same tendency to reduced variation with increasing repetitions. More variation was seen on hard surface (p<0.05) and on the circle (p<0.01). Furthermore, head swivel angle showed a tendency to more variation at recheck compared to day one and day two. Body tracking showed the same tendency to reduced variation with increasing repetitions. More variation was seen at recheck (p<0.05).

For the back segments, there was also a tendency to reduced variation with increased repetitions, but not for T12 flexion-extension, T12 lateral bending, T18 flexion-extension, L3 flexion-extension, L5 lateral bending and lumbosacral lateral bending. More variation at recheck was significant for all segments (p<0.05).

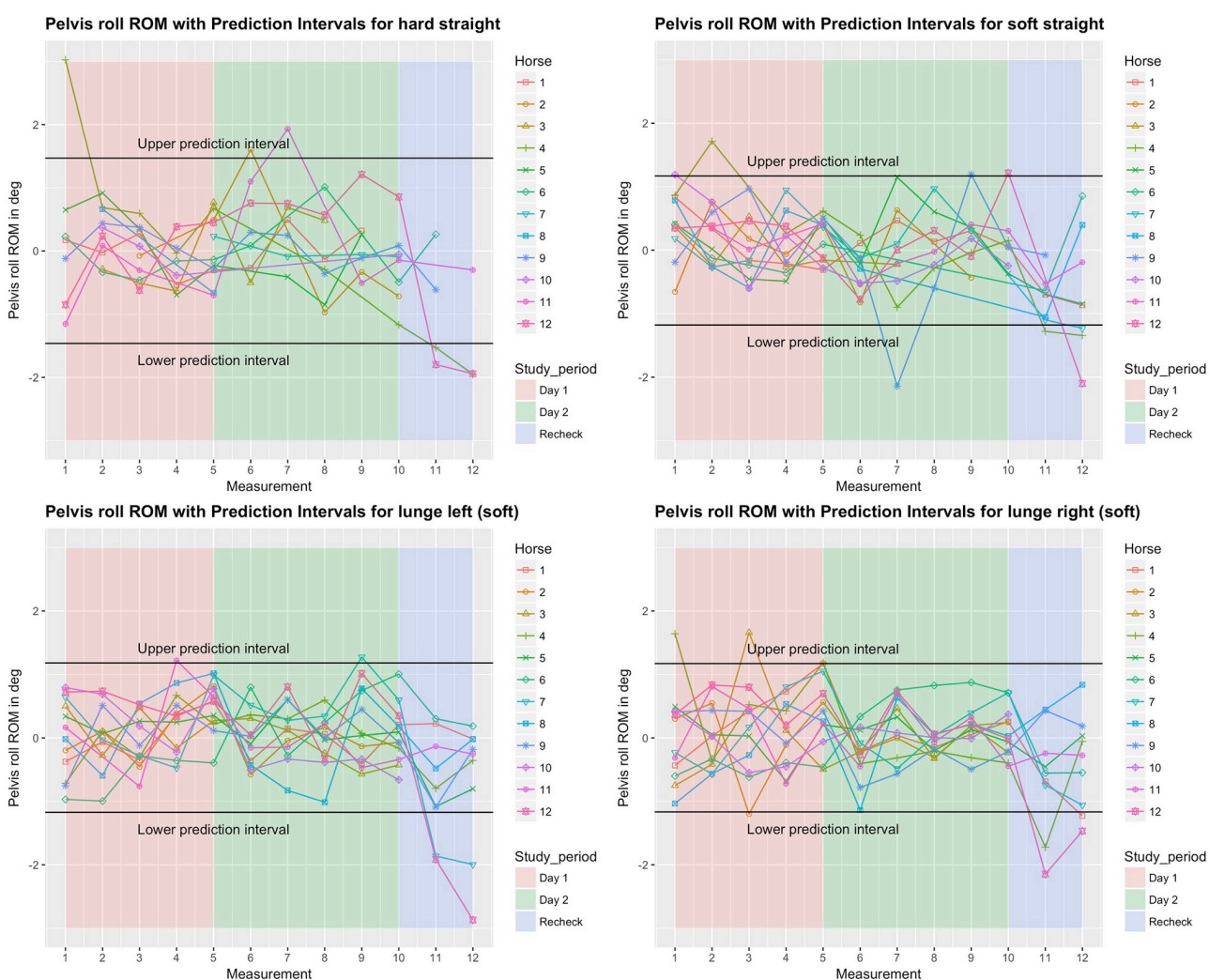

**Fig 5. Between-measurement variation (offset adjusted data) for 'Pelvis roll' (axial rotation of the pelvis) per measurement, per day and per horse (measurement-mean data).** Black lines indicate 95% prediction intervals.

### Intra-class correlation coefficient (ICC)

Table 3 gives an overview of ICC values. Green color-coding indicates the highest ICC values (*i.e.* a better repeatability of these parameters); yellow and red coding indicate moderate and low repeatability, respectively. ICCs were high for pelvic rotations (roll, pitch and yaw), with values ranging from 0.76 and 0.93. For the whole back, ICCs were 0.80–0.91 for lateral bending, and 0.51–0.83 for flexion-extension. ICCs were lower (orange to red scaling) for head swivel (0.22–0.77) and for body tracking (0.62–0.80). Repeatability for the back segments ranged between 0.34 and 0.89. ICCs on the hard, straight line were overall lower compared to all paths on soft surface.

## Discussion

In the present study, ROM and between-measurement variation was investigated for spinal kinematics, measured by optical motion capture. The primary aim was to establish normal ranges for spinal kinematics in clinically sound horses trotting over-ground, which would be

**Fig 6. Between-measurement variation (offset adjusted data) for 'Pelvis pitch' (flexion-extension of the pelvis), per measurement, per day and per horse (measurement-mean data).** Black lines indicate 95% prediction intervals.

useful for comparing conditions before and after intervention or for distinguishing between normal and abnormal movement in horses with suspected back dysfunction.

Although this group of horses (n = 12) is relatively small, differences between horses in back and pelvic ROM were substantial. The 5–95% percentile range corresponds to 30–50% of the stride ROM for the five main parameters (Table 1). Variation in back ROM between horses under comparable conditions can be related to several factors. Conformation, discipline and age have been shown to influence back ROM [14,21]. Higher movement quality, as judged at official performance tests, has been shown to correlate with limb kinematics, for example a longer stride duration, a larger positive diagonal advanced placement and more flexion in the elbow, carpus, hock and hind fetlock joints [26] and could therefore also be an influencing factor for spinal biomechanics. Substantial between-horse variation has been found for lameness parameters as well, indicative of individual motion patterns. [18].

Horses included in this study were perceived sound by their owners, in regular work and scored as sound or less than 1 out of 5 lame on the AAEP scale[22]. Hence, not all horses showed perfect symmetry at trot. This is representative for the sports horse population at large.

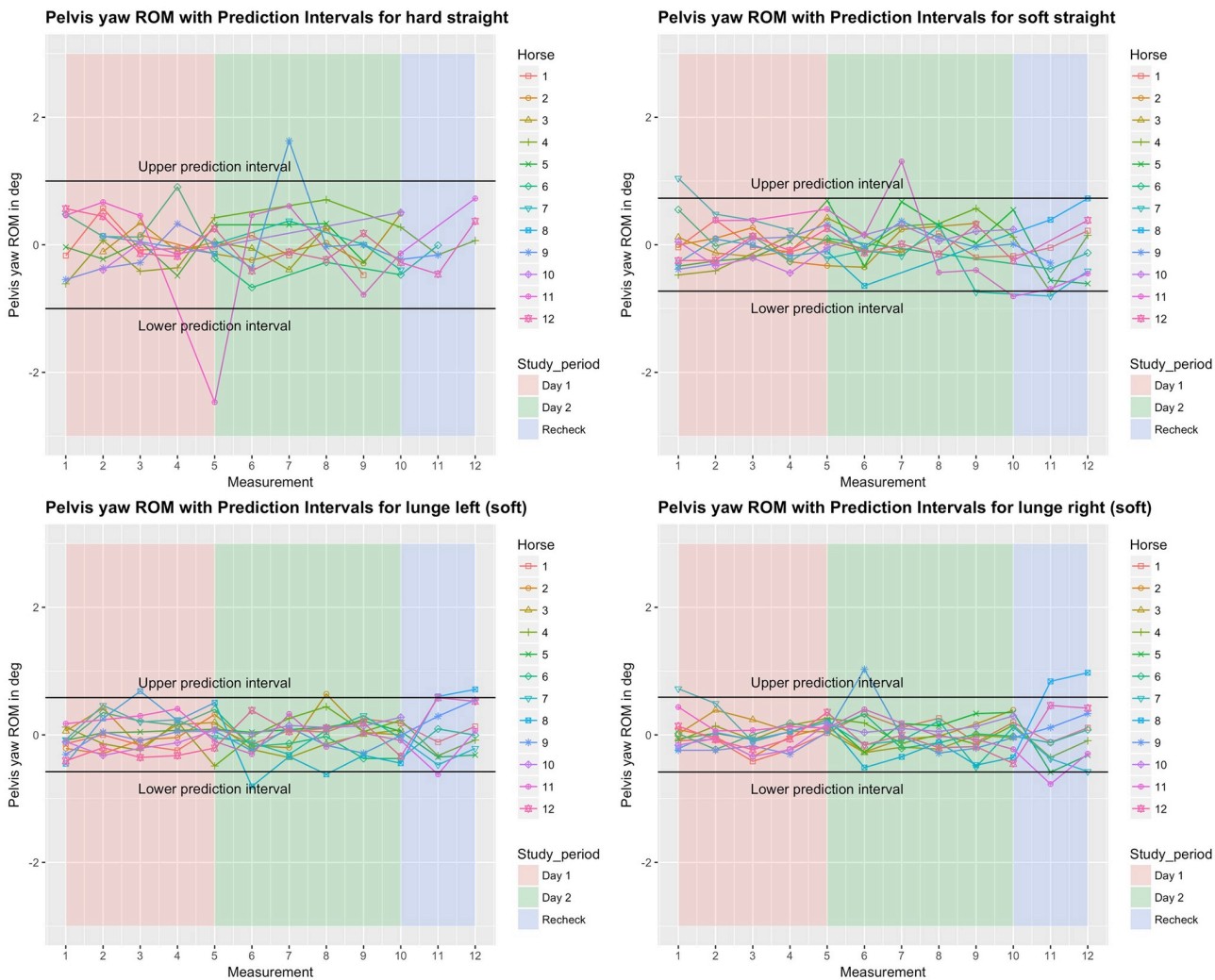

**Fig 7. Between-measurement variation (offset adjusted data) for 'Pelvis yaw' (lateral bending of the pelvis), per measurement, per day and per horse (measurement-mean data).** Black lines indicate 95% prediction intervals.

Earlier studies have shown that a significant proportion of the sports horse population is not classified as completely sound or symmetrical in their gait patterns, irrespective whether assessment is done subjectively by an experienced clinician [27] or evaluated by objective quantitative techniques [28]. It is not known whether symmetry in lameness parameters correlate with back ROM in sound or well-performing horses, but if so, this could be an additional source of between-horse variation.

For head swivel (Fig 2), most horses showed left lateral bending on both hard and soft straight lines (Table 1). This is likely to some extent related to the handler guiding the horse from the left side. However, on the circle most horses also showed considerably more bending to the left on the left circle, compared to right bending on the right circle. It has been discussed whether sidedness in horses, as in this asymmetric cervical bending, is a consequence of human handling or related to innate laterality [29]. Variation in sidedness patterns between horses could influence back ROM, perhaps particularly on the circle. Body tracking (Fig 2) is almost symmetric when comparing left and right circles, and was generally straight on straight

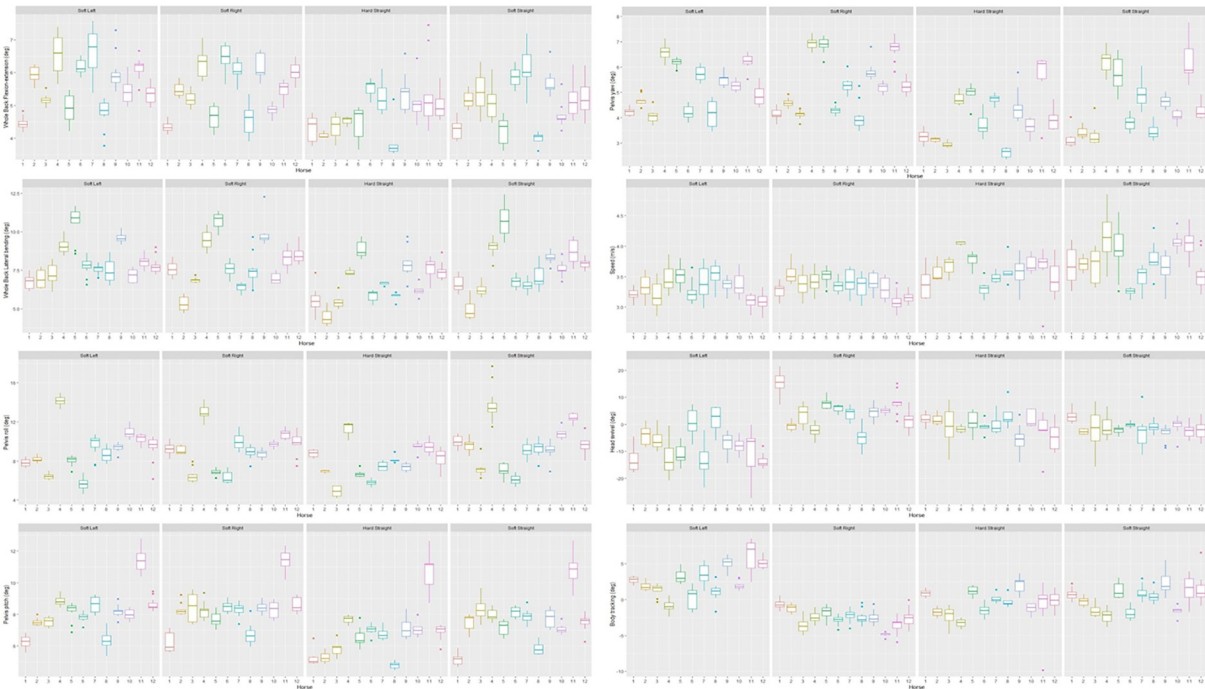

**Fig 8. Between-measurement-variation (Non offset adjusted data) per horse and per path over all measurements.** Here for the parameters 'Whole Back Flexion-extension', 'Whole Back Lateral bending', 'Pelvis roll', 'Pelvis pitch', 'Pelvis yaw', 'Speed', 'Head swivel' and 'Body tracking'. These illustrations enable the evaluation of the absolute values and the differences between versus within horses (relatively small individual boxes compared to the more substantial difference between the different boxplots).

lines, so cervical lateral bending asymmetries appear to be relatively independent from body tracking.

In line with our hypothesis, there was larger between-horse variation compared to within-horse variation. Therefore, measurements of back ROM are clinically more useful if measurements before and after intervention are performed, with the horse being used as its own control. A larger between-horse variation in spinal kinematics, compared to within-horse variation, was observed in a previous study [14]. However, the expected effect size for

**Table 2. Between-measurement variation, given as the (absolute) 95% prediction interval, per condition and per parameter.** Within brackets the between-measurement variation as percentage of the stride ROM for each of the five main parameter. Calculated arithmetic means of the predictions are shown in the last column.

|  | hard straight | soft straight | soft left | soft right | mean variation |
|---|---|---|---|---|---|
| **Flexion-extension** (deg) | 1(20%) | 0.7(14%) | 0.7(12%) | 0.6(11%) | **0.8** |
| **Lateral bending** (deg) | 1.2(19%) | 1.1(15%) | 0.9(12%) | 0.9(12%) | **1** |
| **Pelvis roll** (deg) | 1.5(18%) | 1.2(12%) | 1.2(14%) | 1.2(13%) | **1.3** |
| **Pelvis pitch** (deg) | 1.3(19%) | 0.9(12%) | 0.8(10%) | 1.1(13%) | **1** |
| **Pelvis yaw** (deg) | 1(25%) | 0.7(16%) | 0.6(11%) | 0.6(11%) | **0.7** |
| Head swivel (deg) | 9.8 | 7.1 | 10.1 | 5.7 | **8.2** |
| Body tracking (deg) | 3 | 2.5 | 2.2 | 1.8 | **2.4** |
| Speed (m/s) | 0.6 | 0.5 | 0.3 | 0.3 | **0.4** |

Values are calculated over all 12 horses and all repetitions per horse for each path and surface combination. 'Flexion-extension' and 'Lateral bending' were calculated as the angle between the two segments 'withers—T15' and 'T15—tuber sacrale', in the sagittal plane for flexion-extension and in the dorsal (horizontal) plane for lateral bending.

**Table 3. ICC outcomes.**

|  | Hard straight | Soft straight | Soft left | Soft right |
|---|---|---|---|---|
| Pelvis roll (Axial rotation) | 0.82 | 0.91 | 0.92 | 0.9 |
| Pelvis pitch (Flexion-extension) | 0.86 | 0.9 | 0.9 | 0.84 |
| Pelvis yaw (Lateral bending) | 0.76 | 0.9 | 0.9 | 0.93 |
| Speed | 0.25 | 0.53 | 0.38 | 0.38 |
| Head swivel | 0.23 | 0.22 | 0.59 | 0.77 |
| Body tracking | 0.54 | 0.63 | 0.8 | 0.62 |
| Whole back Flexion-extension | 0.51 | 0.8 | 0.83 | 0.83 |
| Whole back Lateral bending | 0.8 | 0.88 | 0.87 | 0.91 |
| Flexion-extension T12 | 0.83 | 0.87 | 0.86 | 0.85 |
| Lateral bending T12 | 0.78 | 0.78 | 0.82 | 0.8 |
| Flexion-extension T15 | 0.49 | 0.7 | 0.65 | 0.78 |
| Lateral bending T15 | 0.77 | 0.81 | 0.83 | 0.82 |
| Flexion-extension T18 | 0.43 | 0.73 | 0.76 | 0.52 |
| Lateral bending T18 | 0.83 | 0.82 | 0.89 | 0.84 |
| Flexion-extension L3 | 0.64 | 0.84 | 0.76 | 0.86 |
| Lateral bending L3 | 0.68 | 0.85 | 0.78 | 0.87 |
| Flexion-extension L5 | 0.83 | 0.83 | 0.85 | 0.83 |
| Lateral bending L5 | 0.56 | 0.81 | 0.84 | 0.82 |
| Flexion-extension tuber sacrale | 0.34 | 0.53 | 0.54 | 0.38 |
| Lateral bending tuber sacrale | 0.46 | 0.72 | 0.75 | 0.74 |
|  | **0.6205** | **0.753** | **0.776** | **0.765** |

Color coding from red (0.22, lowest values) to green (0.93, highest values).

interventions as mentioned above still needs to be larger than the between-measurement variation. A study comparing spinal kinematics in normal, well-performing horses and horses diagnosed with back pain found rather small differences, comparing them to our prediction intervals of normal variation. In trot, differences in ROM of 0.61 degrees (T17, flexion-extension) and 0.52 degrees (L1, flexion-extension) were found [3]. When horses before and after chiropractic intervention were compared [15], average improvements of 0.3 degrees (T13), 0.8 degrees (T17) for flexion-extension and 0.5 degrees (L3) for lateral bending were found. Comparing this to our results, it turns out that the prediction intervals for between-measurement variation are larger; values of 0.6 to 1.2 degrees in the segmental calculations (S2 Table) and 0.7 to 1.3 degrees for the five main parameters (Table 2). Due to the higher between-measurement variation in our study compared to the differences between symptomatic and asymptomatic, or the differences before and after intervention, objective measurements of back ROM will have inadequate sensitivity for detecting these differences in individual horses.

For most of the studied variables, significant differences in between-measurement variation were found depending on surface and path, with more variation on hard surface for almost all variables, and more variation on the circle for head swivel. There are several explanations for the tendency to more between-measurement variation on the hard surface. First, the shorter trot-up (40 m versus 70 m on the soft surface) implies less strides collected and thereby more influence of single strides on the mean value. Furthermore, the fact that ROM was lower on hard surface compared to soft surface in most horses (Table 1),results in small variations, or any measurement errors, being a larger part of the ROM and consequently in lower ICC values (Table 3). Soft surface reduces impact peak loading and maximal ground reaction forces [30–32], which may make horses feel more comfortable, thereby resulting in a higher ROM. In

human runners, an increased ROM of the pelvis was found on soft surface as well[33]. There is also a possibility that (subclinical) gait irregularities became more manifest on the hard surface. To summarize, taking care to collect enough strides to ensure correct interpretation is important and soft surface is possibly more suited for the assessment of spinal kinematics due to the higher ROM.

The higher variation of the head swivel angle on the circle (compared to the soft straight line) is likely due to more freedom of cervical motion on the lunge. In general, the horses also showed increased back ROM on the circle compared to the straight (Table 1), which is in line with previous findings[12]. This stresses the importance of assessing spinal kinematics on the circle in addition to straight line, but differences in spinal biomechanics between circle and straight line warrant further investigation.

As for the lameness parameters in the earlier study[18], there is a tendency for all five main parameters to reduced variation with increased repetitions. However, a significantly larger difference from the mean of all 12 repeats was seen at recheck (M11-M12, p<0.001). We assume that there is a training effect which makes horses more accustomed to the environment after a few trot-ups, despite a prior warm-up. By the time of the recheck (which included only two measurement), this effect will have worn off. This implicates that, in a clinical situation, the horse should be given enough time to get accustomed to the environment, in order to perform a proper subjective and objective evaluation of locomotion.

Apart from the systematic factors and natural movement variability, between-measurement variation may also have been influenced by issues related to data collection and data quality. Marker placement plays an important role when using optical motion capture and the influence of incorrect marker placement is large when measuring spinal kinematics, because of small ROM; a small misplacement can have significant influence on the results[34]. Marker placement is likely partly responsible for the higher variation at recheck in this study. It will also be difficult to avoid some inconsistency in marker placement in the clinical situation, where one is normally not allowed to clip or mark horses for repeated measurements.

Correcting for speed in our models had minimal influence on the estimates for between-measurement variation. This is a clinically important finding, as it indicates that, when taking the usual care to keep speed as constant as possible, there is no need in a clinical setting to correct for small differences in speed between measurements, for example before and after an intervention.

The ICCs are highest in pelvic motion (Table 3). This can be explained by the pelvis behaving as a rigid body[35], whereas the back segments include anatomical locations containing various joints. Furthermore, marker configuration may play a role here; both tuber coxae and tuber sacrale markers form one single unit and are hence less prone to effects of marker (mis) placement [34]. Repeatability of the whole back flexion-extension and lateral bending is fairly good and similar for the different path and surface combinations (0.80–0.91), except for the hard, straight line in flexion-extension, where ICC is 0.51. Three studies have evaluated between-measurement ICCs for lameness parameters. Using data collected at the same occasion as the data used in this study, ICC values of 0.90–0.99 were found[18]. In thoroughbreds in training, with data collected with IMUs, ICC values ranged from 0.40 to 0.92 across parameters for daily repeats and 0.27 to 0.91 for weekly repeats [19]. Another study using an IMU-based gait analysis system found that same day repeats resulted in ICC values $\geq$0.89 for head vertical movement and $\geq$0.93 for pelvic vertical movement[20]. ICCs have not been previously published for spinal kinematics, but a study on repeatability of back ROM found that variation between horses was at least twice as large compared to variation between days, when quantified as coefficient of variation [14]. In the clinical situation, our results indicate that repeated measurements are reliable for whole back flexion-extension and lateroflexion and for pelvic

roll, pitch and yaw. Concerning the back segments, one should interpret differences before and after a given intervention with more care, as ICC's are clearly lower and hence less reliable. For all parameters, the horse should serve as its own control due to the larger between-horse variation.

The clinical examination of the equine spine is described as subjective and variation exists in the approach to this examination, depending on experience, tradition and personal bias [11]. During lameness assessment, different professionals look at different parameters [36], and the same is likely true for the back. Additionally, the human eye may not be capable of appreciating the small variations in movement symmetry [37], or discriminate between normal and pathological back movement. Preliminary data on agreement between veterinarians/physiotherapists assessing spinal motion showed very poor interclass correlations (T. Spoormakers, personal communication), suggesting potential benefits for evaluating back kinematics objectively. However, our results indicate that solely relying on measurements of back ROM, might not be an effective approach for the objective quantification of back dysfunction. The patterns of the different variables over a stride (Fig 2) and the symmetry of movements, may turn out to be clinically more relevant. Since the movement pattern and ROM of the back differ between gaits, evaluating the horse also in walk and/or canter could add further information to the picture. As pattern recognition is a key capability of the human brain (cerebellum) [38] and some of this capacity can be simulated through machine learning [39,40], there might be future possibilities upcoming, using machine learning to objectively assess spinal biomechanics. Therefore, more research and collaboration between veterinarians, chiropractors, engineers and specialists in the field of objective gait analysis is likely needed to develop clinically applicable methods to improve the quality of evaluation of horses presented for disorders of the neck, back and pelvis.

This study has several limitations. The study was performed on a small population including horses from different disciplines, ages and levels. Before inclusion horses were only evaluated on soft surface, which is uncommon in clinical practice. The correlations between whole back and segment variables were not investigated. It is evident from Table 1 that adding all segments gives a larger ROM than the corresponding whole back variable. These discrepancies are likely due to the fact that the whole back angle approximates back movement as if occurring at a single joint at T15 whereas the segments represent the movement with greater resolution. Also, the mean of the 12 repeats will be more influenced by day 1 and 2 (both 5 repetitions), compared to the recheck, with only 2 repetitions.

## Conclusion

In line with previous findings, variation in back ROM between horses was larger than within horses. However, the between-measurement variation found in the present study was larger compared to reported differences between horses with and without back pain. Optical motion capture is also sensitive to marker misplacement. Combined interpretation of measurements under several conditions, e.g. straight/circle, walk/trot, and assessments of stride patterns (instead of only calculating ROM and minima/maxima) over multiple variables may be a way to increase usefulness of objective measurements of spinal kinematics. Further research and collaboration between experts in several fields is needed to find useful tools and protocols for back evaluation in equine patients.

## Supporting information

**S1 Fig. Between-measurement variation (offset adjusted data) for 'Whole Back Flexion-extension', 'Whole Back Lateral bending', 'Pelvis roll', 'Pelvis pitch', 'Pelvis yaw', 'Speed',**

**'Head swivel' and 'Body tracking'.** These data enable the evaluation of the amount and differences in variation.
(DOCX)

**S1 Table. Time schedule of all measurements (M1-M12).** *Horses were measured at different timepoints during the recheck (M11). ** M12 was done 5 minutes after M11.
(DOCX)

**S2 Table. Between-measurement variation of the back segments in degrees, given as the (absolute) prediction interval, per condition and per parameter.** Calculated arithmetic means of the predictions are shown in the last column.
(DOCX)

**S3 Table. Model estimates 'Variability Model', testing the effect of time, surface and path.** Intercept = referenced level (day one, straight line, soft surface). Significance codes: $0 - < 0.001$ '***' $0.001 - < 0.01$ '**' $0.01 - < 0.05$ '*' $0.05 - < 0.1$ '.'.
(DOCX)

**S4 Table. Raw data as used for the graphical and statistical analysis.**
(XLSX)

## Acknowledgments

The authors would like to sincerely thank the owners of the horses and the staff of 'Tierklinik Luesche' for their assistance.

## Author Contributions

**Conceptualization:** A. M. Hardeman, J. H. Swagemakers, P. R. van Weeren.

**Data curation:** A. M. Hardeman, A. Byström, J. H. Swagemakers, F. M. Serra Bragança.

**Formal analysis:** A. M. Hardeman, A. Byström, L. Roepstorff, F. M. Serra Bragança.

**Investigation:** A. M. Hardeman.

**Methodology:** A. M. Hardeman, A. Byström, P. R. van Weeren, F. M. Serra Bragança.

**Project administration:** A. M. Hardeman.

**Software:** F. M. Serra Bragança.

**Supervision:** A. Byström, L. Roepstorff.

**Visualization:** A. M. Hardeman.

**Writing – original draft:** A. M. Hardeman.

**Writing – review & editing:** A. M. Hardeman, A. Byström, L. Roepstorff, J. H. Swagemakers, P. R. van Weeren, F. M. Serra Bragança.

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
