## [Decision Letter · Decision Letter 0]

12 Nov 2019

PONE-D-19-24742

Range of motion and between-measurement variation of spinal kinematics in sound horses at trot on the straight line and on the lunge.

PLOS ONE

Dear Mrs. Hardeman,

Thank you for submitting your manuscript to PLOS ONE. After careful consideration, we feel that it has merit but does not fully meet PLOS ONE’s publication criteria as it currently stands. Therefore, we invite you to submit a revised version of the manuscript that addresses the points raised during the review process.

We would appreciate receiving your revised manuscript by Dec 27 2019 11:59PM. To enhance the reproducibility of your results, we recommend that if applicable you deposit your laboratory protocols in protocols.io, where a protocol can be assigned its own identifier (DOI) such that it can be cited independently in the future. For instructions see: http://journals.plos.org/plosone/s/submission-guidelines#loc-laboratory-protocols

We look forward to receiving your revised manuscript.

Kind regards,

Chris Rogers

Academic Editor

PLOS ONE

Journal Requirements:

1. Thank you for providing your Ethics statement to state that approval was not needed in this case. We would usually expect approval if the research involves manipulation for the specific purpose of the study. In this case, the manipulation is not routine, so we would typically expect approval. Can you please provide the specific regulations that stipulate that approval is not needed in this case?

2. Please provide further information on how the participating horses were recruited for this study, and in what context recruitment was taking place.

3. Thank you for including your competing interests statement; "The authors have declared that no competing interests exist."

We note that one or more of the authors are employed by a commercial company:

"Tierklinik Luesche GmbH, Luesche, Germany"

Additional Editor Comments (if provided):

Thank you for your submission. Both reviewers have suggested minor revision. Please address these suggestions and submit a revised manuscript.

Reviewers' comments:

Reviewer's Responses to Questions

**Comments to the Author**

1. Is the manuscript technically sound, and do the data support the conclusions?

Reviewer #1: Yes

Reviewer #2: Yes

2. Has the statistical analysis been performed appropriately and rigorously? 

Reviewer #1: Yes

Reviewer #2: I Don't Know

3. Have the authors made all data underlying the findings in their manuscript fully available?

Reviewer #1: No

Reviewer #2: Yes

4. Is the manuscript presented in an intelligible fashion and written in standard English?

Reviewer #1: Yes

Reviewer #2: Yes

5. Review Comments to the Author

Reviewer #1: This is a well written manuscript using a robust study design and appropriate methods to address the function and ROM of normal pelvic and spinal kinematics in horses during overground locomotion. As this information is not currently available in the literature the study makes a valuable contribution to knowledge and will be an important reference for veterinary researchers and practitioners. I have some minor comments:

Line 87: Although this has been published previously I feel it would be useful to include further details here, as EVJ is not open access and also as your findings have large between subject variation.

Line 173 (table 1): It needs to be clear that these are values offset from the mean for each horse.

Table 1: Clarify what you mean by T12 etc. in the legend.

Line 213-217: Although this is detailed the exact number of horses, surfaces, path and days used in the analysis is not clear. Line 227 says all 12 horses over all 12 repetitions, but if M11 and 12 were missing for 3 horses this does not seem possible. Please clarify.

Line 234: Suggest you remove between, i.e. Between-measurement variation……

Line 237: Why is S4 supplementary? I feel these data should be included in the main body of the paper.

Table 2: Provide a more detailed legend for the table which also clearly states that these are ‘whole body’ measurements.

Line 308: I think the manuscripts lacks (either in the intro or in the discussion or both) a description of ‘normal function’ in a straight line and on the lunge. What is the role of the spine and pelvis in normal trotting?

Line 313: Please expand here.

Line 326-335: I thought these data were normalized in Table 1? If so, this may be just more head movement compared to the mean, but the mean may be overall more to the right? I think you need to include S4 data from absolute values when discussing laterality. Body tracking is not symmetric from absolute values. Looks like more bias of forehand right-pelvis left on left circle, but for the other 3 surfaces/paths the opposite is probably true, but to a lesser extent.

General comment: A discussion of back function in trot compared to walk or canter would be helpful. Is the back supposed to stabilize in trot? Also, body to pelvic motion, was this normalized? If so, how did this affect symmetry/laterality?

Although there is a large amount of graphical data in the manuscript and supplementary information it appears that the authors have not included ‘raw’ data values from which the graphical and statistical analysis were created/analysed. This needs to be made available in line with the policy of the journal.

Reviewer #2: PONE D 19 24742 Review

General comments

The stated aim of this study was to quantify in an objective manner the spinal kinematics and the variation appreciated between different measurement times, different surfaces and different days in “owner-sound” horses. This study uses data collected within a larger project for which some results have already been published relating to variation in gait parameters that are measured during lameness evaluation. The manuscript presented here investigates spinal movement, in particular range of motion along the axial skeleton in the same horses measured using state of the art motion capture technology. Overall, the manuscript is well written and well presented.

There is no clear hypothesis stated regarding the authors expectations regarding outcomes of the study and would be good to add a hypothesis/hypotheses that can be commented on in the discussion.

The material and methods are well described and are clear. Where necessary, the authors have referenced previous studies in which certain analysis techniques have been established which is helpful and results in a relatively concise M&M section for a detailed study. How does calibration residual of 3.2 mm relate to an angle measurement of 1 degree, since findings measured in degrees? Is this acceptable?

Statistical analysis – appears to be robust but I am not a statistician so consultation with one might be useful, particularly the paragraph regarding the mixed models.

The graphs that present the prediction intervals with the different measurements on each day and the different days are a good representation of the overall data sets. The authors have found a useful way of presenting the data so the reader (and researcher) can assess visually the large amount of data collected and processed in this study.

The discussion covers the important points but lacked clear relation of the findings to an overall picture of the evaluation of the equine spine and how the findings not only support/refute previous literature but also how they could influence clinical evaluation of the equine spine. More author input/clear comment is required regarding what the findings mean in the context of subjective and or objective evaluation of the equine spine kinematics.

As the manuscript is reviewed, please make sure it is always clear where the topic is within horse variation and where between horse variation. Mostly this is clear but would be worth a careful recheck.

Specific comments

Line Abstract

37 Use “in conclusion”..

38 Is it only subjective, but also objective examination/interpretation that is difficult?

Introduction

42 Since neck motion is also included, is it back pain or axial skeleton pain?

Would “Back pain resulting in movement dysfunction – or something like that work better here?

“which can” to replace “and it can”

45-46 This sounds like it is the rider that has reluctance to bend.. etc. Reword so clear that it is the horse that is showing the signs the rider appreciates

53- Be clearer false positive or false negative diagnosis of back pain …

54 Should be “a” not “an”

58 Greater than what? greater than the subjectivity of lameness evaluation? Say clearly

Use “more subtle” instead or “much subtler”; and be specific about changes in what? Changes in back movement? Response to palpation? All clinical signs of back pain?

73 Aims are clear. No hypothesis given.

Materials and Methods

91-93 Needs to be clearer here. Fit to compete based on lameness assessment, with fitness to compete defined as….. Basically mimicking the FEI trot-up? Or alter wording to match that of your EVJ publication which is clearer.

100 Consider “on subsequent examinations” rather than following days.

103 To the right and left (rather than either side)

104 Delete “respectively”, as not correctly used here. This isn’t very clear. Would it work to say T-shaped strip extending between the t coxae with the “T” at the t sacrale? Can you include a figure for the marker placement?

115 Does this mean all video cameras synchronised during data collection? If yes, then maybe say this earlier in paragraph when describing cameras.

124 Is this a break from measurement or rest from any riding exercise, competition or turnout?

148 , 322 Suggest “performed” rather than “done”

165 The cervical lateral bending seems to be measuring the angle that exists between the trunk and the cervical region with the head at the farthest extend. Is this really cervical bending, since the angle of bend along the neck is not being calculated. Consider if another term for this measurement might be a more accurate description?

196 Consider adding .day 2 (6-10), and 11-12 on day three. That is how these trials are referred to in previous work, so might be more consistent to maintain that here?

198-200 This sentence regarding the testing of the “speed” variable is confusing. Please clarify.

202 , 336 Sentences should not be started with “because” if avoidable. Would “however” work here?

227 Consider adding except for the excluded data sets as mentioned above? Since only 9 horses for M11 and 12 and 61 trials excluded for too few steps.

Table 2 Prediction interval is useful but should this be in the context of the overall variation for each parameter to provide perspective. i.e., if PI is 1 degree but the mean angle range for the selected ROM parameter is 10 this is different than if the angle range is 5 for the selected parameter. I realise that including the other values might make the table more complicated but consider if this would provide necessary context for the reader.

276 Significantly more compared to? And for which parameters?

276 Not clear to what “this” refers. Please use the subject of the sentence – is it the variation that was higher. This sentence is confusing so please restate to make it clearer.

278 Do you mean that speed did not have a significant effect on the model outcomes?

278-280 Again, please clarify - a tendency to what?

281 Suggest - Head swivel angle – for clarity

282 Again, for clarity add… more variation in head swivel angle… and more than what?

286-288 Please list the segments that did not have reduced variation.

Then, reduced variation was noted with increased repetitions for back segment angles on the hard surface.

296-297 Could you state which highest and lowes

Table 3 Check that column headers are correct. All other tables have Hard Straight first then soft straight, but here you have reversed them. Suggest switching order for consistency. Then also visual inspection of the table clearly shows poorer ICC fo hard straight compared to others just based on the colours seen.

Discussion

310-311 Is this 30-50% of the total ROM range of values? Maybe add as a clarification.

I think your comment here links to my comments on Table 2 in terms of putting the PI’s into context of overall ranges for each variable. That will provide a better chance for the reader to appreciate this comment you are making in the discussion.

Since table 1 has more than 5 parameters listed, would it be worth putting them in bold so that reader can easily return to the table and be clear which parameters are the main ones?

313-315 Movement quality is a poorly defined term… is there another way to say what you mean? Is that related to stride length, foot flight, -- and are there certain characteristics of movement quality that would include certain parameters measured in this study?

Individual patterns of what? Be clearer what you are trying to convey to the reader.

316-319 Much of this is repeating M&M so could you simply say – “our inclusion criteria were designed to be representative of the sport horse population that is likely to undergo evaluation of spinal movement as part of an examination? Or something along those lines which would emphasize your reason for the inclusion criteria?

321 Symmetrical in what – their gait patterns?

326 Throughout discussion.. is repeated mention of figures needed? Consider deleting.

327 ? instead of could be…. Use likely related to (if you think that is really the cause)

331 Or is a consequence…

Can you work into this paragraph how cervical bending would effect back ROM. Would it be expected to affect one parameter more than another (pitch, yaw, roll, etc).. so relate your interpretation to the clinical parameter you are measuring. Do you have a suggestion about how to possibly overcome this variation in your data? Randomise and lead from right 50% of time in next study?

336 Try “Due to” in place of “because”

338 Being used as,,,, or serving as its own control

338-341 Expand on this for clarity… within your study, are the within horse variations smaller than between horse.. State clearly as this is the reason for doing the study… (and this could form one of your hypotheses). Were the within horse differences in the quoted study more of less than the between horses? Tie this study in more clearly to your study and your interpretation.

Does “rather small” mean not significant differences… better to just say that for clarity.

345 Between measurement within horse? Maybe worth specifying.

346 Larger than what?

357-359 Double check this reasoning for signal to noise and ICC?

361 Suggest delete “and” and place “,”

363 Can you draw together your statements for this paragraph and maybe say if you think one surface is better and provides mor consistent clinical data than another? Should evaluation of the spine in motion only be done on a soft surface? And ditto for the next paragraph.

368 -373 Sentence awkward? Revise please to first reflect the message you wan to convey (? Learning of horse, warming up?) then link to previous study, then say why it is important to the interpretation of your data and study design.

374 Could you more clearly link this paragraph’s message to the previous – citing marker placement variation to the differences between days 1 2 and recheck?

388 Not sure “regard” is the best word here. ? include?

389-402 Iimportant information as it relates to repeatability and correlation but can you be more succinct and also be more specific about the relevance. In line 395 – tie together the lameness parameters/ICC to the back and why he differences might be important to clinical interpretation. Similar for the IMU statements.. relate to your study and how the other findings support/refute and their relevance to clinical evaluation.

404 Is “variable way” best? Considerable variation exists in the approach to examination of the equine thoracolumbar /cervical region.

406 Suggested rephrase: Additionally, the human eye may not be capable of appreciating the small variations in normal or asymmetric back movement.

409 What is This here/ the preliminary data, the poor agreement… Are there other examples of poor agreement in observation of lameness that could support this statement

428-430 Confusing sentence as now you talk about recheck before day1 and 2. Can you turn sentence around? Should your statistical analysis have accounted for the lost trials? If so, then maybe don’t need to include here.

438 Remove the brackets and rephrase.. The optical motion capture method….

441 Suggest delete “forward”

6. PLOS authors have the option to publish the peer review history of their article (what does this mean?). If published, this will include your full peer review and any attached files.

Reviewer #1: Yes: Dr Sarah Jane Hobbs

Reviewer #2: Yes: Ellen Singer

---

## [Author Response · Author response to Decision Letter 0]

27 Dec 2019

PONE-D-19-24742

Range of motion and between-measurement variation of spinal kinematics in sound horses at trot on the straight line and on the lunge.

PLOS ONE

Dear Mrs. Hardeman,

Thank you for submitting your manuscript to PLOS ONE. After careful consideration, we feel that it has merit but does not fully meet PLOS ONE’s publication criteria as it currently stands. Therefore, we invite you to submit a revised version of the manuscript that addresses the points raised during the review process.

We would appreciate receiving your revised manuscript by Dec 27 2019 11:59PM. To enhance the reproducibility of your results, we recommend that if applicable you deposit your laboratory protocols in protocols.io, where a protocol can be assigned its own identifier (DOI) such that it can be cited independently in the future. For instructions see: http://journals.plos.org/plosone/s/submission-guidelines#loc-laboratory-protocols

• A rebuttal letter that responds to each point raised by the academic editor and reviewer(s). This letter should be uploaded as separate file and labeled 'Response to Reviewers'.

• A marked-up copy of your manuscript that highlights changes made to the original version. This file should be uploaded as separate file and labeled 'Revised Manuscript with Track Changes'.

• An unmarked version of your revised paper without tracked changes. This file should be uploaded as separate file and labeled 'Manuscript'.

We look forward to receiving your revised manuscript.

Kind regards,

Chris Rogers

Academic Editor

PLOS ONE

Journal Requirements:

1. Thank you for providing your Ethics statement to state that approval was not needed in this case. We would usually expect approval if the research involves manipulation for the specific purpose of the study. In this case, the manipulation is not routine, so we would typically expect approval. Can you please provide the specific regulations that stipulate that approval is not needed in this case?

Answer: Under the current legislation on animal experimentation in The Netherlands (“Wet op de Dierproeven” or Act on Animal Experimentation that dates from 1977, but has been changed various times after that) the definition of animal experiments has changed and does not include anymore all manipulations of animals within an experimental setting, but only those causing discomfort above a certain level. The text (Paragraph 1, article 1) stipulates the following as definition of an animal experiment:

Procedure: any use, invasive or non-invasive, of an animal for experimental or other scientific purposes, with known or unknown outcome, or educational purposes, which may cause the animal a level of pain, suffering, distress or lasting harm equivalent to, or higher than, that caused by the introduction of a needle in accordance with good veterinary practice.

This means that procedures causing less discomfort than stated above are not regarded as an experimental procedure on animals. In practice, this means that in case we suppose we are doing experiments we think might fall under this rule (and hence do not need permission), as was the case in the study reported in this paper, we contact our legal adviser on animal experiments and ask him the question. If he confirms our idea, we do evidently not ask permission. Of course, we still need to ask for owner’s consent, which was done.

2. Please provide further information on how the participating horses were recruited for this study, and in what context recruitment was taking place.

Answer: These horses were all owned by veterinarians or technicians from our clinic. They all participated on voluntary basis.

3. Thank you for including your competing interests statement; "The authors have declared that no competing interests exist."

We note that one or more of the authors are employed by a commercial company:

"Tierklinik Luesche GmbH, Luesche, Germany"

Answer: Data collection took place at ‘Tierklinik Lüsche GmbH, Germany’. The first and fourth author of this paper are clinicians working at this clinic, also doing research, which implicates they did play a role in the study design, data collection and analysis, the decision to publish the results and the preparation of the manuscript but without conflicting and/or commercial interests.

Answer: “The funder provided support in the form of salaries for authors [JH (fourth author), AH (first author)], but did not have any additional role in the study design, data collection and analysis, decision to publish, or preparation of the manuscript that could lead to conflicting situations. The specific roles of these authors are articulated in the ‘author contributions’ section.”

Answer: There is no Competing Interest from ‘Tierklinik Lüsche GmbH’ in this study design, study results or decision of publication, neither with related employment, consultancy, patents, products in development or marketed products.

Answer: ‘Tierklinik Lüsche GmbH’ does not alter our adherence to PLOS ONE policies on sharing data and materials.

Additional Editor Comments (if provided):

Thank you for your submission. Both reviewers have suggested minor revision. Please address these suggestions and submit a revised manuscript.

Reviewers' comments:

Reviewer's Responses to Questions

Comments to the Author

1. Is the manuscript technically sound, and do the data support the conclusions?

Reviewer #1: Yes

Reviewer #2: Yes

2. Has the statistical analysis been performed appropriately and rigorously?

Reviewer #1: Yes

Reviewer #2: I Don't Know

3. Have the authors made all data underlying the findings in their manuscript fully available?

Reviewer #1: No

Reviewer #2: Yes

4. Is the manuscript presented in an intelligible fashion and written in standard English?

Reviewer #1: Yes

Reviewer #2: Yes

5. Review Comments to the Author

Reviewer #1: This is a well written manuscript using a robust study design and appropriate methods to address the function and ROM of normal pelvic and spinal kinematics in horses during overground locomotion. As this information is not currently available in the literature the study makes a valuable contribution to knowledge and will be an important reference for veterinary researchers and practitioners. I have some minor comments:

Line 87: Although this has been published previously I feel it would be useful to include further details here, as EVJ is not open access and also as your findings have large between subject variation.

Answer: The information is now added to the manuscript (line 87-96).

Line 173 (table 1): It needs to be clear that these are values offset from the mean for each horse.

Answer: these data are ROM values, using the measurement mean values.

Body tracking and head swivel are the only variables where symmetry can be assessed in Table 1, but these variables must be interpreted with some care in individual horses because of possible asymmetry in marker placement. 

It was added in the legend and in the text to clarify.

Table 1: Clarify what you mean by T12 etc. in the legend.

Answer: This was added to the legend.

Line 213-217: Although this is detailed the exact number of horses, surfaces, path and days used in the analysis is not clear. Line 227 says all 12 horses over all 12 repetitions, but if M11 and 12 were missing for 3 horses this does not seem possible. Please clarify.

Answer: Thanks for noticing, this is indeed not correct in line 227. We changed it to ‘available’ repetitions, as described in detail in line 213-217. 

Line 234: Suggest you remove between, i.e. Between-measurement variation……

Answer: This was removed.

Line 237: Why is S4 supplementary? I feel these data should be included in the main body of the paper.

Answer: We moved S4 to the main document as you suggested and left S5 in the Supplementary Material.

Table 2: Provide a more detailed legend for the table which also clearly states that these are ‘whole body’ measurements.

Answer: The legend was revised with clear, detailed information about the calculations.

Line 308: I think the manuscripts lacks (either in the intro or in the discussion or both) a description of ‘normal function’ in a straight line and on the lunge. What is the role of the spine and pelvis in normal trotting?

Answer: A description of normal back function and the role of the spine was added in the introduction.

Line 313: Please expand here.

Answer: Sorry, but we don’t understand this comment. Please explain if the comment still applies.

Line 326-335: I thought these data were normalized in Table 1? If so, this may be just more head movement compared to the mean, but the mean may be overall more to the right? I think you need to include S4 data from absolute values when discussing laterality. Body tracking is not symmetric from absolute values. Looks like more bias of forehand right-pelvis left on left circle, but for the other 3 surfaces/paths the opposite is probably true, but to a lesser extent.

Answer: Answer: these data are ROM values, using the measurement mean values.. It was added in the legend and in the text to clarify. 

Only for body tracking and head swivel, the direction is given as either plus or minus. For body tracking, a positive value indicates tracking of the forehand to the right and the hind quarters to left. For head swivel, a positive value indicates cervical bending to the right. (line 195-197).

S4 is included in the main document now.

General comment: A discussion of back function in trot compared to walk or canter would be helpful. Is the back supposed to stabilize in trot? Also, body to pelvic motion, was this normalized? If so, how did this affect symmetry/laterality?

Answer: A description on back function was added to the introduction as you suggested. Ranges of motion are not affected by offset adjustment. Body tracking and head swivel are the only variables where symmetry can be assessed in Table 1, but these variables must be interpreted with some care in individual horses because of possible asymmetries in marker placement. 

Although there is a large amount of graphical data in the manuscript and supplementary information it appears that the authors have not included ‘raw’ data values from which the graphical and statistical analysis were created/analysed. This needs to be made available in line with the policy of the journal.

Answer: We added all the data used in the analysis and for graphics in the Supplementary Material. We referred to this at the beginning of the results section.

Reviewer #2: PONE D 19 24742 Review

General comments

The stated aim of this study was to quantify in an objective manner the spinal kinematics and the variation appreciated between different measurement times, different surfaces and different days in “owner-sound” horses. This study uses data collected within a larger project for which some results have already been published relating to variation in gait parameters that are measured during lameness evaluation. The manuscript presented here investigates spinal movement, in particular range of motion along the axial skeleton in the same horses measured using state of the art motion capture technology. Overall, the manuscript is well written and well presented.

There is no clear hypothesis stated regarding the authors expectations regarding outcomes of the study and would be good to add a hypothesis/hypotheses that can be commented on in the discussion.

Answer: The hypothesis has been added at the end of the introduction. 

The material and methods are well described and are clear. Where necessary, the authors have referenced previous studies in which certain analysis techniques have been established which is helpful and results in a relatively concise M&M section for a detailed study. 

How does calibration residual of 3.2 mm relate to an angle measurement of 1 degree, since findings measured in degrees? Is this acceptable?

Answer: 3.2 mm corresponds to 1.2 degrees for a segment of the length 0.15 m (relevant for segments) and 0.4 degrees for a segment of the length 0.5 m (relevant for all other variables). For the segments this is a rather large error, but this is for any given frame and when doing a mean value over several strides the effect is reduced because measurement error is randomly distributed. The calibration residual (3.2 mm) is within the range that you will typically get with an optical motion capture system. 

Still, the residual does not per say represents an estimation error of 3.2 mm. This is simply an average ‘residual’ after interpolation of the position of the marker, when several cameras track the same marker. This inaccuracy is also further improved by filtering, a crucial step when processing motion capture data. 

We do agree that the segments are thereby less repeatable, which we also discussed in the paper. However, we have clarified what we meant by the signal to noise ratio. 

Statistical analysis – appears to be robust but I am not a statistician so consultation with one might be useful, particularly the paragraph regarding the mixed models.

The graphs that present the prediction intervals with the different measurements on each day and the different days are a good representation of the overall data sets. The authors have found a useful way of presenting the data so the reader (and researcher) can assess visually the large amount of data collected and processed in this study.

The discussion covers the important points but lacked clear relation of the findings to an overall picture of the evaluation of the equine spine and how the findings not only support/refute previous literature but also how they could influence clinical evaluation of the equine spine. More author input/clear comment is required regarding what the findings mean in the context of subjective and or objective evaluation of the equine spine kinematics.

Answer: Multiple small changes have been made throughout the discussion, also based on the other reviewer’s comments. We hope that this is satisfactory. 

As the manuscript is reviewed, please make sure it is always clear where the topic is within horse variation and where between horse variation. Mostly this is clear but would be worth a careful recheck.

Answer: This has been checked through the manuscript and we have made the distinction clearer in a few places where this was indeed necessary. 

Specific comments

Line Abstract

37 Use “in conclusion”..

Answer: This has been changed. 

38 Is it only subjective, but also objective examination/interpretation that is difficult?

Answer: Both are difficult in our opinion, as addressed in the discussion. We changed it in line 38 as well as this is indeed an important, clinically relevant, conclusion. 

Introduction

42 Since neck motion is also included, is it back pain or axial skeleton pain?

Would “Back pain resulting in movement dysfunction – or something like that work better here?

“which can” to replace “and it can”

Answer: Neck motion is indeed included in our study, but not in the studies referred to in here. Therefore, we would like to keep it as back pain.

‘and it can’ is replaced by ‘which can’.

45-46 This sounds like it is the rider that has reluctance to bend.. etc. Reword so clear that it is the horse that is showing the signs the rider appreciates

Answer: This is indeed a bit confusing. We changed this according to your request.

53- Be clearer false positive or false negative diagnosis of back pain …

Answer: this was changed to false positive, based on the study referred to (11).

54 Should be “a” not “an”

Answer: Changed accordingly.

58 Greater than what? greater than the subjectivity of lameness evaluation? Say clearly

Answer: This was changed accordingly. 

Use “more subtle” instead or “much subtler”; and be specific about changes in what? Changes in back movement? Response to palpation? All clinical signs of back pain?

Answer: This was corrected and specified in ‘changes in ROM’. 

73 Aims are clear. No hypothesis given.

Answer: The hypothesis has been added directly after the aims of the study. 

Materials and Methods

91-93 Needs to be clearer here. Fit to compete based on lameness assessment, with fitness to compete defined as….. Basically mimicking the FEI trot-up? Or alter wording to match that of your EVJ publication which is clearer.

Answer: It is indeed mimicking the FEI trot-up. Words were changed to ‘sound or close to sound’ as used in the EVJ publication. We kept in the term ‘fit to compete’ to make it clearer for the readers (i.e. clinicians).

100 Consider “on subsequent examinations” rather than following days.

Answer: we would like to keep it like this as it should be clear that markers were not removed between all subsequent examinations, but only between the days. 

103 To the right and left (rather than either side)

Answer: This was changed accordingly. 

104 Delete “respectively”, as not correctly used here. This isn’t very clear. Would it work to say T-shaped strip extending between the t coxae with the “T” at the t sacrale? Can you include a figure for the marker placement?

Answer: This was changed accordingly. We will include a figure as we did in the EVJ publication to visualize marker placement.

115 Does this mean all video cameras synchronised during data collection? If yes, then maybe say this earlier in paragraph when describing cameras.

Answer: This is only about the (single) videocamera, not about the infrared cameras. We rephrased to make this clear.

124 Is this a break from measurement or rest from any riding exercise, competition or turnout?

Answer: This was a break from the measurement. The text was changed accordingly to make it clearer. 

148 , 322 Suggest “performed” rather than “done”

Answer: This was changed as you suggested. 

165 The cervical lateral bending seems to be measuring the angle that exists between the trunk and the cervical region with the head at the farthest extend. Is this really cervical bending, since the angle of bend along the neck is not being calculated. Consider if another term for this measurement might be a more accurate description?

Answer: This is correct. We have changed it to ‘approximating cervical bending’. 

196 Consider adding .day 2 (6-10), and 11-12 on day three. That is how these trials are referred to in previous work, so might be more consistent to maintain that here?

Answer: we changed it as you suggested. 

198-200 This sentence regarding the testing of the “speed” variable is confusing. Please clarify.

Answer: We agree on your comment. We have rephrased the sentence to make it clearer.

202 , 336 Sentences should not be started with “because” if avoidable. Would “however” work here?

Answer: It was changed to ‘as’, which suits better in our opinion than ‘however’. 

227 Consider adding except for the excluded data sets as mentioned above? Since only 9 horses for M11 and 12 and 61 trials excluded for too few steps.

Answer: This was changed as you suggested. 

Table 2 Prediction interval is useful but should this be in the context of the overall variation for each parameter to provide perspective. i.e., if PI is 1 degree but the mean angle range for the selected ROM parameter is 10 this is different than if the angle range is 5 for the selected parameter. I realise that including the other values might make the table more complicated but consider if this would provide necessary context for the reader.

Answer: We like your comment and agree that this is a very useful addition. We added within brackets after each PI the percentage of the stride ROM (out of Table 1). 

276 Significantly more compared to? And for which parameters?

Answer: From the model, horses show more variation on the hard straight line compared to the soft straight line, as is this the path where everything is compared to. In line 286, it is said that this is about the five main parameters (Figs 2-6).

276 Not clear to what “this” refers. Please use the subject of the sentence – is it the variation that was higher. This sentence is confusing so please restate to make it clearer.

Answer: ‘This’ refers to significantly more variation at the recheck. It was changed as you suggested. 

278 Do you mean that speed did not have a significant effect on the model outcomes?

Answer: Yes, that is true. We rephrased the sentence to make this clearer.

278-280 Again, please clarify - a tendency to what?

Answer: A tendency of speed to have an effect on the model outcomes (p=0.08). This was changed in the text. 

281 Suggest - Head swivel angle – for clarity

Answer: This was changed as you suggested. 

282 Again, for clarity add… more variation in head swivel angle… and more than what?

Answer: This was added. Also, ‘compared to day one and day two’ was added. 

286-288 Please list the segments that did not have reduced variation.

Answer: The segments that did not have reduced variation were: T12 flexion-extension, T12 lateral bending, T18 flexion-extension, L3 flexion-extension, L5 lateral bending and tuber sacrale lateral bending. This was added to the text. 

296-297 Could you state which highest and lowes

Answer: this was added to the legend. 

Table 3 Check that column headers are correct. All other tables have Hard Straight first then soft straight, but here you have reversed them. Suggest switching order for consistency. Then also visual inspection of the table clearly shows poorer ICC fo hard straight compared to others just based on the colors seen.

Answer: We switched hard and soft straight for consistency as you suggested. Indeed, ICCs are overall lower on the hard straight line. This is mentioned in the text, visualised in the color scaling and illustrated by the lower overall ICC at the end of the column. 

Discussion

310-311 Is this 30-50% of the total ROM range of values? Maybe add as a clarification.

Answer: Yes, that is correct. This was added to the text. 

I think your comment here links to my comments on Table 2 in terms of putting the PI’s into context of overall ranges for each variable. That will provide a better chance for the reader to appreciate this comment you are making in the discussion.

Answer: This was changed according to your comment above. 

Since table 1 has more than 5 parameters listed, would it be worth putting them in bold so that reader can easily return to the table and be clear which parameters are the main ones?

Answer: That makes it indeed much easier to identify the main parameters. This was changed. 

313-315 Movement quality is a poorly defined term… is there another way to say what you mean? Is that related to stride length, foot flight, -- and are there certain characteristics of movement quality that would include certain parameters measured in this study?

Answer: This is indeed poorly defined but this terminology was taken from the paper which we refer to; movement quality, as judged at official performance tests. This was added to the text, including some of the objective parameters that were found to be significantly different between the two groups. 

For this study, we can only hypothesize on what parameters would perhaps correlate to the subjectively judged ‘movement quality’ of the back. Possibly movement symmetry and, related to today’s the dressage horses nowadays, a high degree of flexion-extension (especially in the lumbosacral junction) and lateral bending and still a longer stride duration, as found in the research referred to. This would be a nice theme for further research. 

Individual patterns of what? Be clearer what you are trying to convey to the reader.

Answer: This is indeed not clear in the text. It was corrected. 

316-319 Much of this is repeating M&M so could you simply say – “our inclusion criteria were designed to be representative of the sport horse population that is likely to undergo evaluation of spinal movement as part of an examination? Or something along those lines which would emphasize your reason for the inclusion criteria?

Answer: We shortened the description as there was indeed some repetition. We did not want to change to a population as presented to the clinic for the evaluation of spinal movement as that was not the selection criterium for our study population. 

321 Symmetrical in what – their gait patterns?

Answer: Exactly. This was added to the text. 

326 Throughout discussion.. is repeated mention of figures needed? Consider deleting.

Answer: We appreciate your comment. We have considered this but feel that repeated mentioning of the figures doesn’t disturb and may be helpful for the reader.

327 ? instead of could be…. Use likely related to (if you think that is really the cause)

Answer: We think it is only to some extent related to handling from the left as the same effect is still seen on the lunge. This was corrected in the text. 

331 Or is a consequence…

Can you work into this paragraph how cervical bending would effect back ROM. Would it be expected to affect one parameter more than another (pitch, yaw, roll, etc).. so relate your interpretation to the clinical parameter you are measuring.

Answer: This was changed to ‘consequence of’. The correlation between variables is outside the scope of this paper. We have added that further research is needed, but this sentence is in a different section (Discussion, line 402).

Do you have a suggestion about how to possibly overcome this variation in your data? Randomise and lead from right 50% of time in next study?

Answer: We did a pilot in handling the horses from the right, which made us decide not to use this in our study because the horses were completely confused by it, also after multiple trot-ups. Furthermore, it does not mimic the clinical situation. Randomisation is indeed a useful possibility for further research, although we doubt whether it would make a difference in this study design as left and right circle were always performed directly after each other, with only the time to turn in between. 

336 Try “Due to” in place of “because”

Answer: This was corrected as you suggested. 

338 Being used as,,,, or serving as its own control

Answer: This was corrected as you suggested.

338-341 Expand on this for clarity… within your study, are the within horse variations smaller than between horse.. State clearly as this is the reason for doing the study… (and this could form one of your hypotheses). 

Answer: The text was expanded so hopefully it now clearly states these findings.

Were the within horse differences in the quoted study more of less than the between horses? Tie this study in more clearly to your study and your interpretation.

Answer: The study of Wennerstrand et al. does not compare within versus between horse differences, neither did the study of Gomez Alvarez et al., comparing horses before and after chiropractic manipulation. 

Does “rather small” mean not significant differences… better to just say that for clarity.

Answer: This were significant differences in the study of Wennerstrand et al. The differences are rather small in relation to our prediction intervals of normal variation of the same spinal locations. This was added in the text. 

345 Between measurement within horse? Maybe worth specifying.

Answer: This between-measurement variation is calculated over the whole group as the variation that can be expected when doing repeated measurements within and over multiple days of the same horse. For this reason, it would not be correct to specify as ‘within horse’ as the variation was calculated over the whole group. 

346 Larger than what?

Answer: larger than the value found in the cited study as clarified in the sentence before. 

357-359 Double check this reasoning for signal to noise and ICC?

Answer: We have tried to express clearer in this sentence what we mean.

361 Suggest delete “and” and place “,”

Answer: This was changed as you suggested. 

363 Can you draw together your statements for this paragraph and maybe say if you think one surface is better and provides mor consistent clinical data than another? Should evaluation of the spine in motion only be done on a soft surface? 

Answer: A short resume at the end of the paragraph was added. We don’t think assessment should only be done on soft surface, but due to the higher ROM it is possibly a more suitable surface than a hard surface. 

And ditto for the next paragraph.

Answer: Forthis paragraph also a short resume was added. 

368 -373 Sentence awkward? Revise please to first reflect the message you wan to convey (? Learning of horse, warming up?) then link to previous study, then say why it is important to the interpretation of your data and study design.

Answer: This sentence was corrected and the importance of the interpretation of our data for the clinical situation was added. 

374 Could you more clearly link this paragraph’s message to the previous – citing marker placement variation to the differences between days 1 2 and recheck?

Answer: We don’t exactly understand your comment as this is discussed a bit further on in the manuscript. We are happy to make more changes if you still feel that this is needed.

388 Not sure “regard” is the best word here. ? include?

Answer: ‘Regard’ was replaced by ‘include’.

389-402 Iimportant information as it relates to repeatability and correlation but can you be more succinct and also be more specific about the relevance. In line 395 – tie together the lameness parameters/ICC to the back and why he differences might be important to clinical interpretation. Similar for the IMU statements.. relate to your study and how the other findings support/refute and their relevance to clinical evaluation.

Answer: A resuming sentence was added to highlight the clinical implication of the results found.

404 Is “variable way” best? Considerable variation exists in the approach to examination of the equine thoracolumbar /cervical region.

Answer: This is indeed a better way of phrasing. It was changed according to your suggestion.

406 Suggested rephrase: Additionally, the human eye may not be capable of appreciating the small variations in normal or asymmetric back movement.

Answer: The sentence was rephrased in line with your suggestion.

409 What is This here/ the preliminary data, the poor agreement… Are there other examples of poor agreement in observation of lameness that could support this statement

Answer: The preliminary data show poor agreement. The sentence was optimized to make this clearer. Unfortunately, there are, as far as we know, no studies done on the subjective agreement of clinicians on spinal motion in horses.

428-430 Confusing sentence as now you talk about recheck before day1 and 2. Can you turn sentence around? Should your statistical analysis have accounted for the lost trials? If so, then maybe don’t need to include here.

Answer: The sentence was corrected to clarify. The sentence about the lost trials was deleted. 

438 Remove the brackets and rephrase.. The optical motion capture method….

Answer: This was corrected as you suggested.

441 Suggest delete “forward”

Answer: This was deleted as you suggested.

6. PLOS authors have the option to publish the peer review history of their article (what does this mean?). If published, this will include your full peer review and any attached files.

Do you want your identity to be public for this peer review? For information about this choice, including consent withdrawal, please see our Privacy Policy.

Reviewer #1: Yes: Dr Sarah Jane Hobbs

Reviewer #2: Yes: Ellen Singer

---

## [Editor Report · Decision Letter 1]

29 Jan 2020

Range of motion and between-measurement variation of spinal kinematics in sound horses at trot on the straight line and on the lunge.

PONE-D-19-24742R1

Dear Dr. Hardeman,

We are pleased to inform you that your manuscript has been judged scientifically suitable for publication and will be formally accepted for publication once it complies with all outstanding technical requirements.

With kind regards,

Chris Rogers

Academic Editor

PLOS ONE

Additional Editor Comments (optional):

Thank you for the revised manuscript. You have adequately addressed the comments by the reviewers.
---

## [Editor Report · Acceptance letter]

13 Feb 2020

PONE-D-19-24742R1 

Range of motion and between-measurement variation of spinal kinematics in sound horses at trot on the straight line and on the lunge. 

Dear Dr. Hardeman:

I am pleased to inform you that your manuscript has been deemed suitable for publication in PLOS ONE. Congratulations! Your manuscript is now with our production department. 

With kind regards,

on behalf of

Dr. Chris Rogers 

Academic Editor

PLOS ONE